# Swimming Behavior of *Daphnia magna* Is Altered by Pesticides of Concern, as Components of Agricultural Surface Water and in Acute Exposures

**DOI:** 10.3390/biology12030425

**Published:** 2023-03-10

**Authors:** Nicole Egan, Sarah A. Stinson, Xin Deng, Sharon P. Lawler, Richard E. Connon

**Affiliations:** 1School of Veterinary Medicine, University of California at Davis, Davis, CA 95616, USA; 2California Department of Pesticide Regulation, Sacramento, CA 95812, USA; 3Department of Entomology and Nematology, University of California at Davis, Davis, CA 95616, USA

**Keywords:** photomotor response, chlorantraniliprole, imidacloprid, mixture toxicity, agricultural runoff, first flush, neonicotinoid, aquatic toxicology

## Abstract

**Simple Summary:**

New types of pesticides are increasingly found in surface waters around the world. Little is known about how they interact in mixtures, and how these mixtures might then affect aquatic organisms at risk of exposure. We looked at how two new pesticides, chlorantraniliprole (CHL) and imidacloprid (IMI), affected a sensitive aquatic organism, *Daphnia magna*. We exposed *Daphnia* to surface waters known to be contaminated by agricultural runoff at two time points: during an extended dry period and after the first seasonal storm event. In surface waters, the concentrations of CHL, IMI, and other pesticides of concern increased after the storm event. Exposure to these waters caused *Daphnia* to be hypoactive, and their response to light varied with the concentration of surface water. We then exposed *Daphnia* to each chemical individually and then to mixtures of the two chemicals, at concentrations that occur in polluted waterways. *Daphnia* exposed to CHL and IMI mixtures were more active in response to light stimuli than the control group. *Daphnia* swimming behavior is a sensitive way to measure the biological effects of CHL, IMI, and surface waters.

**Abstract:**

Pesticides with novel modes of action including neonicotinoids and anthranilic diamides are increasingly detected in global surface waters. Little is known about how these pesticides of concern interact in mixtures at environmentally relevant concentrations, a common exposure scenario in waterways impacted by pesticide pollution. We examined effects of chlorantraniliprole (CHL) and imidacloprid (IMI) on the sensitive invertebrate, *Daphnia magna*. Exposures were first performed using surface waters known to be contaminated by agricultural runoff. To evaluate the seasonal variation in chemical concentration and composition of surface waters, we tested surface water samples taken at two time points: during an extended dry period and after a first flush storm event. In surface waters, the concentrations of CHL, IMI, and other pesticides of concern increased after first flush, resulting in hypoactivity and dose-dependent photomotor responses. We then examined mortality and behavior following single and binary chemical mixtures of CHL and IMI. We detected inverse photomotor responses and some evidence of synergistic effects in binary mixture exposures. Taken together, this research demonstrates that CHL, IMI, and contaminated surface waters all cause abnormal swimming behavior in *D. magna.* Invertebrate swimming behavior is a sensitive endpoint for measuring the biological effects of environmental pesticides of concern.

## 1. Introduction

Global waterways are impacted by chemical contaminants including pesticides which can negatively impact sensitive species, and human and environmental health [1]. Pesticides with novel modes of action have become increasingly common as they replace older pesticide classes such as organophosphates [2,3,4]. The use of both chlorantraniliprole (CHL) and imidacloprid (IMI) is widespread and these pesticides of concern are now detected in many surface waters around the world. Chlorantraniliprole detections are frequent in agriculture regions worldwide [5,6,7,8,9]. Similarly, use of neonicotinoids such as IMI has increased in recent times [10]; IMI is one of the most frequently used insecticides across the world [11]. Imidacloprid and CHL are frequently detected environmental pesticides of concern which affect an organism’s nervous system [12,13,14]; however, the extent of their impact on aquatic organisms has not been fully evaluated. These chemicals have the potential to affect an organism’s behavior due to their impacts on important neurological receptors [12,14,15]. Chlorantraniliprole, like other anthranilic diamides, is classified as a Ryanodine Receptor (RyR) modulator, specifically by activating and competing for binding of the RyR [12,16]. This receptor affects behavior by altering calcium signaling and muscle movement [17]. Imidacloprid is a neonicotinoid pesticide which interacts agonistically with the postsynaptic nicotinic acetylcholine receptor (nAChR) causing toxic effects to the central nervous system [14]. Despite their relative reduced environmental persistence [18,19], these chemicals have been shown to cause adverse effects on non-target aquatic organisms, such as sensitive invertebrate species including the model organism, *Daphnia magna* [12,20]. Both of these pesticides have been shown to cause changes in swimming behavior in *D. magna* [21]. While single-chemical exposure effects are well documented for these novel pesticides of concern, little is known about how CHL and IMI interact in mixtures at environmentally relevant concentrations.

An initial seasonal rain event occurring after a period of dry weather, referred to as a “first flush” event, can result in sudden influxes of pesticides into waterways. Runoff or partitioning of water-soluble pesticides into surface waters can affect aquatic health, but certain weather events, like first flush events, can exacerbate this issue. This is especially true in Mediterranean climates, such as California (USA), which are characterized by dry summer months followed by winter rain events. In the absence of irrigation or other mitigating circumstances, this period of little or no rainfall allows pesticide buildup to occur prior to first storm events. First flush events may mobilize pesticides that were applied during extended periods, causing a sudden spike in pesticide concentrations in surrounding waterways [22,23,24,25,26]. Cladocerans, including *Daphnia* spp., are the dominant group of zooplankton in many freshwater waterbodies, both in biomass and abundance, and can have significant effects on aquatic food chains via their role in the regulation of phytoplankton abundance and competition with other zooplankton [27,28]. Disruption of this basal trophic level can also result in the reduction of energy transfer efficiency to predators such as fish [29]. Invertebrates which have bioaccumulated pesticides may represent a greater risk to their predators [30,31]; however, the biotransformation processes of IMI and CHL are largely unknown for aquatic invertebrates.

Survival of model species is a commonly used endpoint for determining the potential toxicity of surface water [32]. This endpoint, however, does not fully capture the adverse effects of chemical exposures, especially when evaluating environmentally relevant concentrations. Behavioral assessments following exposure to sublethal concentrations of pesticides are extremely powerful as they can capture underlying physiological or biochemical conditions, which manifest themselves at the organismal level [33]. This approach can determine ecological risk if the behavior directly relates to factors like survival, growth, or reproduction [34]. Swimming behavioral assays can show adverse effects at much lower chemical concentrations than other commonly measured toxicological endpoints [33], making them useful for analyzing pesticides at levels far below their lethal concentrations [35]. 

Swimming behavior is a well-established endpoint in fish studies for pharmacology and toxicology [1,36,37,38]. However, invertebrate species are generally underrepresented. Certain invertebrates have been shown to be more sensitive than most fishes during toxicity testing and are easily obtained and kept in a laboratory setting [39]. Invertebrate behavioral testing has the potential to become a powerful tool in the field of aquatic toxicology and water quality monitoring [37,40]. *Daphnia magna* have well defined acute toxicity testing parameters [32] and are known to demonstrate measurable changes to their natural swimming behavior in response to pesticide exposure, which can be linked with their overall fitness [41]. Despite these advantages, there are few data evaluating the behavioral effects of complex environmental mixtures on *D. magna,* an important fish prey and indicator species for multiple sensitive aquatic invertebrates. The Salinas Valley (CA, USA) is a highly productive agricultural region which exports a diverse array of agricultural commodities (i.e., lettuces, strawberries, tomatoes) across the world. Due to the intense agricultural activity in this area, complex mixtures of many pesticide classes are frequently detected in runoff. Surface water monitoring has been routinely conducted in this area for more than ten years by the California Department of Pesticide Research (CDPR) and the Central Coast Regional Water Quality Control Board [42]. Analytical chemistry data from these monitoring efforts document frequent detections of many global pesticides of concern, including CHL, IMI, other neonicotinoids, pyrethroids, and other pesticides used in California [13]. Based on this information, we decided to utilize this region as a representative sample of areas with high agricultural use. 

In this study, we aimed to assess the effects of two emerging pesticides of concern, CHL and IMI, two known neurotoxicants that are frequently found in monitored agricultural waterways at levels exceeding the United States Environmental Protection Agency (EPA) benchmarks for aquatic life [12,13,14,43,44,45,46,47,48]. We evaluated the swimming behaviors of *D. magna* as sensitive bioindicators of exposure to a dilution series of surface water samples collected from an agricultural region (Salinas County, CA), during an extended dry period, and after a first flush event. We used a dilution series with surface water concentrations ranging from 100% to 6% in order to observe a wide range of toxicological outcomes. To isolate the effects of these two pesticides from other chemicals present in these mixtures, we also evaluated the survival and swimming behaviors of *D. magna* after acute (96 h) exposures to single and binary mixtures of CHL and IMI, at concentrations relevant to those observed in surface water.

## 2. Materials and Methods

### 2.1. Toxicity Testing

#### 2.1.1. Test Organisms

*Daphnia magna* and their food were obtained from Aquatic Biosystems Inc. (Ft. Collins, CO, USA). Upon arrival, we fed 48 h neonates *ad libitum* with a mixture of suspended *Raphidocelis subcapitata* and YCT (yeast, cerophyl, trout chow mixture). We maintained them at 20 ± 2 °C under a 16 h light:8 h dark photoperiod in EPA synthetic control water [49]. We made synthetic, medium hard control water (hereafter referred to as control water) consisting of deionized water modified with salts to meet EPA freshwater specifications, and which was prepared by dissolving 23.04 g NaHCO_3_, 14.40 g CaSO_4_·2H_2_O, 14.40 g MgSO_4_, and 0.96 g KCl in 120 L of deionized water to achieve a hardness of 160–180 mg/L CaCO_3_ and alkalinity of 110–120 mg/L CaCO_3_.

#### 2.1.2. Acute Exposure Conditions

We performed all exposures following US EPA protocols [49]. For field exposures, we placed twenty individuals into each of the 250 mL replicate beakers containing 200 mL of treatment water, with four replicates per treatment. We used larger exposure volumes for the field water to reduce the potential influence of sediment on organism toxicity, and to follow EPA guidelines for acute exposures to effluent [49]. For CHL and IMI exposures, we exposed six organisms in 20 mL scintillation vials, with six replicates per treatment, per time point. We randomly chose 24 individuals per treatment group to use in behavioral assays. We conducted all exposures in temperature-controlled chambers kept at 20 ± 2 °C, with a 16 h:8 h light:dark cycle to maintain optimal conditions for our test organisms [49]. Every day during the exposure, we recorded the number of organisms per beaker and the mortality, while removing dead individuals from the tests. At the 48 h mark, we performed water changes; 80% volume was exchanged in surface water exposures to account for suspended solids and additional bacterial activity seen in field samples, and 50% water changes for CHL/IMI exposures. We tested temperature, total alkalinity, hardness, pH, and dissolved oxygen in situ using a YSI EXO1 multi-parameter water quality sonde at both test initiation and 48 h to ensure that the water remained within the acceptable ranges for *D. magna.* We fed all organisms at both the test initiation and after 48 h water renewals [49].

### 2.2. Surface Water Sample Collection 

We targeted sampling sites located in the Salinas River and surrounding waterways (Salinas Co., Monterey, CA, USA) which correspond to long-term monitoring sites. Data from these sites include more than 10 years of historical chemical analysis data. Historically, some pesticide detections exceeded EPA Benchmarks for Aquatic Life [13,43]. These sites are also located near ecologically sensitive areas and are thus of interest for monitoring water quality [47,50]. These sampling sites are also located downstream of highly productive agricultural regions and residential areas, leaving them at high risk for contamination of complex mixtures. We sampled surface water before and ~24 h after a first flush event (17 September and 26 November 2019) from these sites. We collected samples from well mixed, wadable waters using 1 L amber glass bottles (Cole-Parmer, Vernon Hills, IL, USA) certified to meet current EPA guidelines, sealed with Teflon-lined lids. Immediately after collection, we placed samples in coolers on wet ice for transportation, then refrigerated them at 4 °C upon arrival in the lab. We initiated all acute exposure tests within 24 h of surface water collection. 

#### Surface Water Exposures Dilution Series

Based on high invertebrate mortality previously observed in water from two of the sites (97.5–100%) [51], we made a dilution series of our water samples to capture a wider range of toxic effects including mortality (lethal) and swimming behavior (sublethal). For before first flush sampling, we used a dilution series of surface water concentrations—100%, 60%, 35%, 20%, and 12%—in order to evaluate the potential for a wide range of toxicological outcomes. We thoroughly mixed ambient surface water samples by agitation immediately before creating the dilutions in order to homogenize the turbidity levels between dilutions. To create the dilution series, we added control water (described in Section 2.1.2) to ambient surface water to achieve each desired concentration. We repeated this procedure at the 48 h point when performing an 80% water change on all treatment groups. For after first flush sampling, we used a broader dilution series—100%, 30%, 20%, 12%, and 6%—in anticipation of higher chemical concentrations based on previous studies [51]. We tested temperature, total alkalinity, hardness, pH, and dissolved oxygen in situ using a YSI EXO1 multi-parameter water quality sonde at both test initiation and 48 h to ensure that the water remained within the acceptable ranges for *D. magna*.

### 2.3. Imidacloprid and Chlorantraniliprole

We chose exposure concentrations of CHL and IMI to mimic environmentally relevant concentrations found in monitored agricultural waterways, as well as experimental EC50/LC50 values [13,44,45,47]. For both CHL and IMI, the low and high concentrations were 1.0 μg/L and 5.0 μg/L, respectively. We purchased chemicals (CHL and IMI) from AccuStandard (New Haven, CT, USA). We dissolved CHL in pesticide grade acetone to make chemical stock solutions, subsequently diluting it with EPA synthetic control water (described in Section 2.1.2) to a final concentration of 0.1 mL/L in exposure water. Due to its solubility, no solvent was needed to make an IMI stock solution [46]. To account for this difference, we compared CHL treatment data to an acetone solvent control, and IMI to the EPA synthetic control water. The California Department of Food and Agriculture Center for Analytical Chemistry analyzed these chemical stock solutions via LC-MS MS. 

### 2.4. Chemical Analyses

Chemical analysis of field water was conducted at the Center for Analytical Chemistry, California Department of Food and Agriculture (Sacramento, CA, USA) using multi-residue liquid chromatography tandem mass spectrometry (LC-MS/MS) and gas chromatography–mass spectrometry (GC-MS/MS) methods. Chemicals were analyzed following procedures described in the Monitoring Prioritization Model as mentioned on the CPDR’s website [47,48]. Chlorantraniliprole and IMI stock solutions were also analyzed to confirm exposure concentrations. The method detection limit and reporting limit for each analyte are listed in Appendix A. Laboratory QA/QC followed CDPR guidelines provided in the Standard Operating Procedure CDPR SOP QAQC012.00 [44]. Extractions included laboratory blanks and matrix spikes.

### 2.5. Behavioral Assays and Data Analysis

We performed behavioral assays at the 96 h time points for both the chemical exposures and for the field sampling exposures. We designed behavioral assays using Ethovision XT™ software (version 14.0; Noldus, Wageningen, The Netherlands), and adjusted the video settings (contrast, light level, etc.) to maximize the software’s detection of *D. magna.* We gently transferred organisms from test vessels into randomized wells in a non-treated 24 round-well cell culture plate (Thermo Fisher Scientific, Cleveland, OH, USA) containing 1 mL of control water at 20 °C. We then left them to habituate for at least one hour before moving them to our behavioral assay set up for an additional five-minute acclimation period. The DanioVision™ Observation Chamber (Noldus Information Technology, Leesburg, VA, USA) had a temperature-controlled water flow-through system, allowing us to keep organisms at optimal temperature throughout the assay. Our CCD video camera recorded the entire plate in which the organisms were held throughout the assay, so in this case 24 individuals were assessed at the same time. Using the Ethovision XT™ software, we then analyzed each video frame identifying the location of the organisms at each time point. Calculations were carried out to produce quantified measurements of the organisms’ behavior including both total distance moved and velocity. This assessment of horizontal movement over time, measured as total distance moved, is useful when trying to determine the changes in locomotor ability of organisms after exposure to pesticides [52,53]. This system also allows us to control the dark:light cycle throughout the assay in order to measure endpoints related to a light stimulus, including photomotor response. We measured significant changes in photomotor responses as the change in mean (±SE) distance traveled between the last 1 min of a light photoperiod and the first minute of the dark photoperiod as described in Steele et al. (2018) [33].

We checked data sets for normality using a Shapiro–Wilk test and applied log transformations before statistical analysis. We used a repeated measure ANOVA to analyze the effects over the light period. Statistical tests were defined by treatment (surface water sample site or IMI/CHL treatment) as between-subject factors, and time (5 min time bins) as the within-subject factor. We applied Dunnett’s multiple comparison test (α-level set at 0.05) for post hoc evaluation [52]. Data are represented as mean ± standard error of the mean (S.E.M.). 

We exported summary statistics from Ethovision XT using 1 min time bins for each treatment and analyzed the data in GraphPad Prism, version 9.0 (GraphPad Software, San Diego, CA, USA). We determined significance of mortality data by Analysis of Variance (ANOVA) followed by Dunnett’s test for multiple comparisons one-way analysis using GraphPad Prism, version 8.0. To measure the photomotor response of the organisms, we calculated the difference in distance moved between the last minute of the dark period and the first minute of the subsequent light period for each individual [38]. These data sets were then log transformed and analyzed in GraphPad Prism using a one-way ANOVA with a Tukey’s Post Hoc test of multiple comparisons.

## 3. Results

### 3.1. Chemical Analyses

#### 3.1.1. Surface Water Analytical Chemistry—Before First Flush (September 2019)

Chemicals detected in the water samples collected in September are shown in Appendix A, and are described in further detail in Stinson et al. 2021, a parallel study. In brief, of 47 pesticides analyzed, 17 were detected in our surface water samples, and each site contained a minimum of 7 target pesticides. Chlorantraniliprole was detected at all sites at concentrations below the acute lethality benchmarks for invertebrate species exposure [43] (LC50 = 7.1 µg/L; EPA benchmark for acute, 5.8 µg/L, and chronic, 4.47 µg/L). The neonicotinoid IMI was detected above the EPA benchmark for chronic invertebrate exposure (0.01 µg/L), and above the acute invertebrate level (0.385 µg/L) at Alisal Creek (0.513 µg/L). Neonicotinoids were detected at all sites. Organophosphates were detected at two of the sites: Quail Creek and Alisal Creek. Several pyrethroids were detected at levels at or above an EPA benchmark, including permethrin, lambda-cyhalothrin, and bifenthrin (analytes of particular concern). Several other chemical detections exceeded EPA benchmark values. Notably, methomyl was detected at Quail Creek (29.9 µg/L) at nearly three times the limit for chronic fish exposure (12 µg/L), and above the EPA benchmark for chronic invertebrate exposure (0.7 µg/L) at all sites. Overall, Salinas River contained the smallest total number of chemicals at the lowest concentrations of the three sites we examined. 

#### 3.1.2. Surface Water Analytical Chemistry—After First Flush (November 2019)

Chemicals detected in water samples collected in November are shown in Appendix A. Of 47 pesticides analyzed, 27 were detected in our surface water samples, and each site contained a minimum of 21 target pesticides. Chlorantraniliprole was detected at all sites below the lowest benchmark (3.02 µg/L). The neonicotinoid IMI was detected above the EPA benchmark for chronic invertebrate exposure (0.01 µg/L) at Salinas River (0.03068 µg/L), Alisal Creek (0.29254 µg/L), and Quail Creek (0.30697 µg/L). Neonicotinoids and organophosphates were detected at all sites. Several pyrethroids were detected at levels at or above an EPA benchmark, including permethrin, cyfluthrin, lambda-cyhalothrin, bifenthrin, fenpropathrin, esfenvalerate (analytes of particular concern). Overall, Salinas River contained the smallest total number of pesticides at the lowest concentrations of the three sites we examined. 

#### 3.1.3. Relative Change of Target Chemicals before and after First Flush

We compared the relative change in water chemistry before and after a first flush rain event for all pesticides that exceeded an EPA benchmark for aquatic life value during at least one sampling event (Appendix A). Analytes which increased in concentration from September to November are shown in yellow and increased values that also exceeded EPA acute invertebrate aquatic life benchmarks are shown in red. Approximately half of the chemicals detected at levels above EPA benchmarks had an increase in concentration after the first flush event at each site, and two thirds of the detected pyrethroid pesticides increased to levels above the EPA acute invertebrate aquatic life benchmarks in all three sites.

#### 3.1.4. Chlorantraniliprole and Imidacloprid 

All test concentrations for CHL and IMI stock solutions were confirmed to be within 15% of the nominal value, and no contamination was detected in either of the control groups (Appendix A).

### 3.2. Mortality and Behavioral Assays

#### 3.2.1. Surface Water Exposures September 2019 (before First Flush Event)

After 96 h exposures (described in Section 2.4), we measured significant mortality for two of the three sites tested, both in 100% field-collected surface water and in the geometric dilutions (Table 1). Physicochemical parameters (temperature, total alkalinity, hardness, pH, and dissolved oxygen) for the exposure period are listed in Appendix A. Exposure to surface water collected from Quail Creek and Alisal Creek resulted in significant mortality in all dilutions and were thus classified as toxic. We observed 100% mortality in *D. magna* exposed to surface water from Quail Creek treatments, even in the lowest concentration (12% surface water). For *D. magna* exposed to surface water from Alisal Creek, we observed 100% mortality in the two highest concentrations (100% and 60%). Therefore, we were unable to run behavioral assays on these treatment groups because we lacked the individuals to do so. As a result, we performed behavioral analysis for the remaining treatment groups from Alisal Creek and Salinas River.

We measured changes in total distance moved and photomotor response from behavioral assays (described in Section 2.5). Repeated measures ANOVA showed there were no time-by-treatment interactions, but there were significant (*p* = 0.0025) effects of treatment, on locomotor activity (Appendix A). *Daphnia magna* exposed to 35% and 20% surface water from Alisal Creek exhibited significantly hypoactivity compared to the control group under light conditions (Figure 1A). Additionally, *D. magna* exposed to 20% surface water from Alisal Creek exhibited significant hypoactivity compared to the control group under dark conditions of the behavioral assay. *Daphnia magna* exposed to the highest concentration of surface water from Alisal Creek tested were significantly hypoactive during the last 5 min of the exposure period. Organisms exposed to all concentrations of surface water from Salinas River were hyperactive under light conditions with the two highest concentrations showing the greatest hyperactivity when compared to controls (Figure 1B). There was no difference in total distance moved (mm) between organisms exposed to the Salinas River dilution series and the control group individuals in the dark period.

The photomotor response for organisms exposed to surface water from both Alisal Creek and Salinas River followed a clear log-linear dose-response curve (Figure 2C and Figure 3D). Both the control and solvent control groups exhibited a reduction in movement consistent with a freeze response. Overall, Alisal Creek exposed organisms showed a greater magnitude of change than Salinas River exposed organisms.

#### 3.2.2. Surface Water Exposures November 2019 (after First Flush)

For surface water samples obtained immediately following the first rain event of the season (first flush event), percentages of mortality are shown below in Table 2. We detected significant mortality in undiluted (100%) surface water from Quail Creek and Alisal Creek, and in the 20% treatment of Alisal Creek. Physicochemical parameters (temperature, total alkalinity, hardness, pH, and dissolved oxygen) for the exposure period are listed in Appendix A.

We measured changes in total distance moved and photomotor response with behavioral assays as described in Section 2.5. Repeated measures ANOVA showed that there was a significant effect of both treatment (*p* < 0.0001) and time period (*p* < 0.0001) on locomotor activity (Appendix A). *Daphnia magna* exposed to surface water from all sites tested had significant changes to behavior when compared with controls, during at least one time period and light condition measured (Figure 2A–C). Organisms exposed to the lowest percentage of surface water from Quail Creek (6%) were significantly hypoactive under both dark and light conditions. Organisms exposed to surface water from all sampling sites were significantly hypoactive in the dark, with the exception of the two highest Quail Creek concentrations tested (12% and 20%). Individuals exposed to the highest percentage of surface water from Alisal Creek were hypoactive for the duration of the light period. Salinas River was the only sampling site with enough survivors to test the behavior of individuals exposed to undiluted surface water. Individuals from this treatment were the most hypoactive when compared with controls. 

There were significant changes in photomotor response across all treatment groups, though responses differed between sampling sites. *Daphnia magna* exposed to water samples from Quail Creek (*p* = 0.0011) demonstrated an inverse dose response pattern, where exposure to the lowest dilution gave the most significant change in photomotor response, and exposure to the highest dilution was not significantly different from control groups (Figure 2D). The Alisal Creek (*p* = 0.0386) treatment groups exhibited a non-monotonic dose response, with organisms exposed to the medium dosage having little to no response to light stimulus. The low (6%) dilution had a significantly lessened photomotor response pattern, and the highest dilution was not significantly different from the control group (Figure 2B). *Daphnia magna* exposed to all concentrations of surface water from Salinas River had significantly altered photomotor responses (*p* = 0.0051) as compared to controls. Organisms exposed to undiluted water samples from Salinas River demonstrated an opposite startle response of equal magnitude to the control’s freeze response.

#### 3.2.3. Chlorantraniliprole and Imidacloprid Exposures

Physicochemical parameters (temperature, total alkalinity, hardness, pH, and dissolved oxygen) for the exposure period are listed in Appendix A. Following 96 h exposures, we measured no significant mortality in *D. magna* after exposure to CHL or IMI, at either the high (5.0 µg/L) or low (1.0 µg/L) concentrations following the 96 h acute exposure period (Appendix A). Repeated measures ANOVA (Appendix A) showed there were no time-by-treatment interactions for any experiment, but there were significant effects of both time and treatment, individually, on locomotor activity in the CHL/IMI data sets (time bin: *p* < 0.0001; treatment: *p* < 0.0001). Both the control and solvent control groups exhibited a large photomotor response consistent with freezing (*p* = 0.0009).

After exposure to the low level of CHL, *D. magna* showed hypoactivity under dark conditions (Figure 3A). For *D. magna* exposed to both low and high treatments of IMI, we saw significant hypoactivity during the entire behavior assay period, under both light and dark conditions (Figure 3B). Exposure to mixtures of CHL and IMI resulted in divergent total distance moved measurements under both light and dark conditions. Individuals from the low CHL/low IMI treatment group were hypoactive in dark conditions. In contrast with the single chemical exposures, individuals from the high CHL/low IMI treatment group were hyperactive under light conditions.

We measured significant changes in photomotor responses between the last 1 min of a light photoperiod and the first minute of the dark photoperiod (*p* = 0.0009). The change in total distance moved (mm) during the dark:light transition is shown in Figure 3D–F.

For both CHL treatments, organisms exhibited no response to light stimulus (shown in Figure 3D), representing a nearly 60-fold difference in response from the control group. Organisms exposed to low IMI had an inverse response to light stimulus when compared to the control group, increasing their total distance moved in response to light stimulus. Organisms exposed to high IMI exhibited a reduction in their average total distance moved, but this response was fivefold smaller than controls. 

Mixtures of CHL and IMI resulted in the most divergent photomotor response, when compared with controls. *Daphnia magna* in all binary treatment groups, with the exception of the low CHL/low IMI group, showed an inverse photomotor response from controls. 

## 4. Discussion

Surface water from all sites contained CHL and IMI as components of complex mixtures (including other neonicotinoids, pyrethroid insecticides, organophosphates, carbamate insecticides, and other pesticides of concern) from surface water at all sites, both before and after a first flush event. Several chemicals detected from these sites are known to have sublethal effects on *D. magna*, including IMI, CHL, bifenthrin, clothianidin, malathion, methomyl, and lambda-cyhalothrin [40,54,55,56,57]. The changes in pesticide composition and concentration between the sampling dates concurred with results from previous chemical analyses in this region [42]. Pesticides of concern including CHL and IMI were detected at higher concentrations after the first flush event (Appendix A). A study examining first flush toxicity in California found that the concentration of pollutants (including pesticides) was between 1.2 and 20 times higher at the start of the rain season versus the end [24]. Interestingly, the sampling site with the highest increase in concentration after first flush, for several pesticides of concern, was the Salinas River site. The Salinas River sampling site has been used as a least-impacted reference site in previous toxicity studies [42,51] and is generally classified as non-toxic, based on acute exposure studies. This increase in potency after a rain event is consistent with an influx of pesticides, and the chemical analyses show higher levels of several pesticides of concern in November at all three sites (Appendix A). Climate change is altering rainfall patterns in many areas of the world and understanding how these changes may impact sensitive aquatic systems is crucial for monitoring water quality [25].

Surface water exposure caused significant changes in *D. magna* swimming behavior both before and after a first flush event, even at low concentrations. In September, prior to first flush, we detected strong dose-response patterns in total distance moved and log-linear dose response in photomotor response. *Daphnia magna* exposed to all concentrations of surface water in September increased their movement in response to light stimulus, while control groups reduced their activity. This may have implications for survival in natural populations. Individuals who cannot respond to predator cues, or that show impaired and/or altered responses, may have an increased risk of predation [58]. It is important to note that changes in swimming behavior in organisms exposed to water samples from Alisal Creek in September may have been partially capturing a lethal response. This treatment group had significant mortality in all exposure concentrations (Table 1), so it is possible these individuals were exhibiting not only sublethal, but also delayed lethal toxic responses. Future studies should consider including recovery periods in their experimental design and analyses to parse out whether behavioral impacts are reversable, indicative of long-term effects, or even subsequent mortality. Due to the high mortality observed for Quail Creek in September, we were unable to make any behavioral comparisons. It is notable that the level of methomyl detected at this site was greater than three times the EPA chronic fish exposure level [43], and it is likely that methomyl represents a main driver of the toxicity for this site. It is possible that additional contaminants are present at this site, which were not included in our analysis. Many pharmaceuticals are known to cause hyperactivity and have been detected in wastewater at other sites in California [59,60,61]. Taken together, these findings illustrate the importance of conducting sublethal assessments to link physiological responses to chemical monitoring data.

After the first flush (November), we measured hypoactivity for all sites during at least one light condition, in at least one concentration. Many of the pesticides we detected in surface water samples are known to reduce the swimming speed and distance of *D. magna* at concentrations relevant to those detected in our samples [57,62,63]. We detected changes to the photomotor responses of *D. magna* exposed to low concentrations of surface water from all three sites when compared with controls, demonstrating biologically relevant impacts. Despite low mortality observed in the Salinas River site during both testing dates, we detected altered behavior even at the highest dilution of 6% ambient water in November. Hypoactivity and altered photomotor responses may reduce the capacity of *D. magna* to follow normal behaviors, such as patterns of diel vertical migration and horizontal distribution, thus increasing predation risk and reducing overall fitness [58,64]. We measured significant changes in the swimming behavior of *D. magna* after acute exposures to CHL and IMI in single and binary chemical exposures, and as components of agricultural surface waters both before and after a first flush event. Surface waters contained complex mixtures including CHL and IMI, but also other pesticides of concern including neonicotinoids, pyrethroids, carbamates, and organophosphates. We determined that swimming behaviors of *D. magna* (measured as average total distance moved and photomotor response) are sensitive endpoints for the sublethal assessments of the tested pesticides, and for surface water exposures. 

We detected chemical-specific changes in *D. magna* swimming behavior for both CHL and IMI exposures. Imidacloprid exposure at environmentally relevant concentrations caused hypoactivity for both concentrations tested, across both dark and light conditions, following a dose-response pattern. The increase in activity over the light period represents a return to baseline following a change in light conditions. Our results are consistent with previous findings: IMI negatively impacts nerve conduction and alters swimming behavior in *D. magna* [54] and is known to inhibit acetylcholinesterase (AChE). Past research has shown AChE inhibition is linked to changes in swimming response, and that a 50% decrease in AChE activity can cause enough change in swimming behavior in *D. magna* to be described as toxic [54]. In a recent study examining the effects of IMI on the amphipod *Gammarus fossarum*, IMI stimulated locomotor activity at low exposure concentrations (0.1 μg/L) and inhibited activity at higher concentrations (1.0 μg/L) [55]. *Daphnia magna* are particularly tolerant to neonicotinoids [14], illustrating the potential for impacts in other more sensitive organisms known to inhabit IMI-polluted waterways. We detected significant hypoactivity in individuals exposed to CHL under dark conditions. This is consistent with previous studies on *D. magna* demonstrating that CHL is a known neurotoxicant for this species, causing changes in muscle contraction via interaction with the ryanodine receptor [12]. Low (μg/L) levels of CHL exposure have been shown to produce dose-dependent inhibition of swimming, and decreased responses to light stimulation in a recent study [53]. Another recent study examining effects of single chemical exposure to CHL and IMI, among other chemicals, at low (1.0 μg/L) concentrations had effects on total distance moved of *D. magna* [21]. 

We observed hypoactivity under dark conditions and hyperactivity under light conditions for *D. magna* after exposure to binary mixtures of CHL and IMI. Hyperactivity could suggest a possible disruption of signal transmission in the vision or nervous systems and has been observed for IMI exposures at low (μg/L) exposure levels in other studies [53]. The hyperactivity observed in the low IMI exposure group was notable in that the response was inverse from both single chemical exposures performed at the same concentrations, potentially indicative of an antagonistic response. Our finding is partially consistent with Hussain et al. (2020) [21]; however, where investigators also found hyperactivity under light conditions but no significance under dark conditions. It is relevant to note that our experimental design differed from that of Hussain et al. (2020), who used one exposure vessel containing 50 Daphnids per treatment group, whereas we used fewer Daphnia (n = 24) per exposure vessel, with six exposure vessels per treatment. For future studies, increased replication could improve the ability to determine whether small changes in total distance moved could also be significant. Considering the significance of our other treatments and endpoints, and that our replication exceeded many previously published studies, we propose that our experimental design was sufficient to detect many sublethal effects [21,56]. 

Sublethal impacts can result in ecologically relevant effects on individual fitness, populations, and communities. In pesticide-contaminated aquatic environments, overall invertebrate biomass and diversity are reduced as sensitive individuals and species decline [20]. With the increasing number of pesticides being detected in waterways worldwide, rapid and standardized testing approaches are urgently needed. For many species and chemicals of interest, biochemical reactions can visually manifest via behavioral changes, making behavior a highly integrative and informative endpoint for exposure [65]. Meta-analysis of behavior in comparison to other toxicological endpoints such as development, lethality, and reproduction, showed that behavioral analyses are advantageous to assess the effects of environmental chemicals due to their relative speed and sensitivity [66]. Behavioral assays possess great potential as rapid, high throughput monitoring tools [1,33].

## 5. Conclusions

To our knowledge, this study is the first to assess the behavioral responses of *D. magna* after exposure to mixtures of CHL and IMI. There is little data on the effects of complex environmental mixtures on *D. magna* swimming behavior. The extensive literature available on their use as sensitive bioindicators for acute single chemical exposure studies provides a solid baseline for future studies on mixture toxicity. Considering that most pesticide exposure events occurring in natural environments involve complex mixtures, more research is needed to assess mixture effects. In addition to CHL and IMI, we detected additional neonicotinoids, pyrethroid insecticides, organophosphates, carbamate insecticides, and other pesticide classes in surface waters collected from an agricultural region (Salinas, CA). The Salinas River site, historically quantified as the least toxic site in this monitoring area, demonstrated the greatest impact to the swimming behavior of *D. magna* during both sampling dates. This finding illustrates the importance of including sublethal endpoints in assessing biological exposure impacts. An assessment exclusively evaluating survival would have concluded that this site had negligible impacts on invertebrate species. These findings support the need for further studies to link changes in swimming behavior to large scale ecological effects. We detected CHL, IMI, additional neonicotinoids, pyrethroid insecticides, organophosphates, carbamate insecticides, and other pesticide classes in surface waters both before (during an extended dry period) and after a first flush event. Changes in chemical detection between sampling dates are consistent with hypothesized first flush event effects, including an increase in concentrations of multiple pesticides. This was especially true of pyrethroid detections at both Alisal Creek and Salinas River sampling sites which saw notable increases in chemical concentrations, which in some cases surpassed the EPA aquatic life benchmark for acute invertebrate exposure. Our findings indicate that swimming behaviors are sensitive endpoints for assessing sublethal effects of exposure to IMI in *D. magna* neonates. We found that acute exposures to CHL and IMI caused changes in swimming behavior of *D. magna,* at environmentally relevant concentrations. Standardization of behavioral assays is also necessary to increasing the replicability and relevance of future work. Sublethal impacts on behavior can result in ecologically relevant effects on individual fitness, populations, and communities via exposure to pesticides, and should be used as a routine endpoint for exposure assessments. Future studies should consider including behavior when assessing the effects of environmental chemicals, for both single chemical assessments and in environmentally relevant mixtures. Swimming behavior is a sensitive endpoint to assess the effects of complex mixtures that may impact freshwater ecosystems across multiple trophic levels.

## Figures and Tables

**Figure 1 biology-12-00425-f001:**
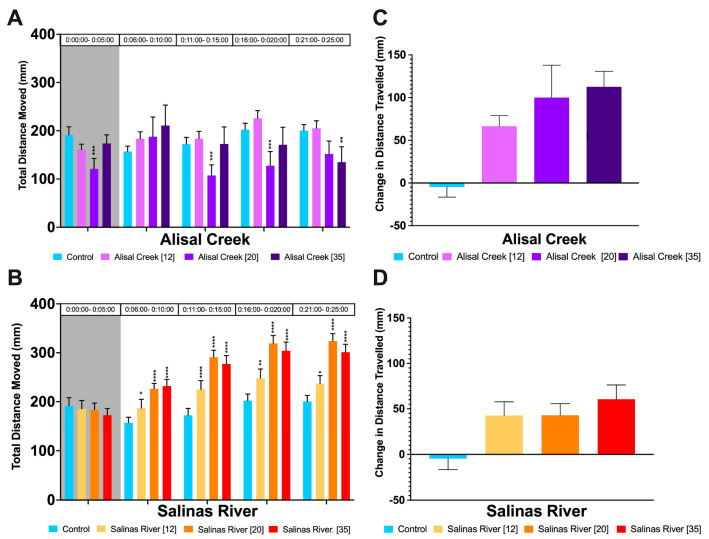
Swimming Behavior of *D. magna* after Exposure to Water Samples from the Salinas Valley Watershed collected in September 2019 (before first flush). (**A**,**B**) Average total distance moved (mm) of *D. magna* after 96 h of exposure to water samples from (**A**) Alisal Creek or (**B**) Salinas River. (**C**,**D**) Photomotor response of *D. magna* after 96 h of exposure to water samples from (**C**) Alisal Creek or (**D**) Salinas River, calculated by taking the difference between the average distance moved in the last minute of dark and the first minute of light. The significance of Tukey’s Multiple Comparisons tests is shown as * = *p* ≤ 0.05, ** = *p* ≤ 0.01, *** = *p* ≤ 0.001, **** = *p* ≤ 0.0001. Error bars depict standard error.

**Figure 2 biology-12-00425-f002:**
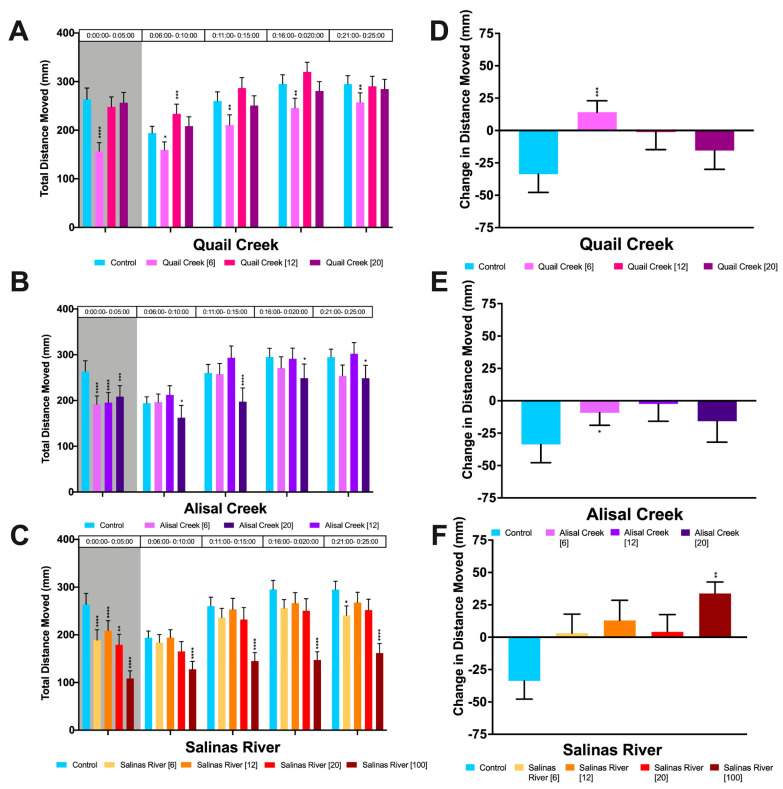
Swimming Behavior of *D. magna* after Exposure to Water Samples from the Salinas Valley Watershed collected in November 2019 (after first flush). (**A**–**C**) Average total distance moved (mm) of *D. magna* after 96 h of exposure to water samples from (**A**) Quail Creek, (**B**) Alisal Creek or (**C**) Salinas River. (**D**–**F**) Photomotor response of *D. magna* after 96 h of exposure to water samples from (**D**) Quail Creek, € Alisal Creek, or (**F**) Salinas River calculated by taking the difference between the average distance moved in the last minute of dark and the first minute of light. The significance of Tukey’s Multiple Comparisons tests is shown as * = *p* ≤ 0.05, ** = *p* ≤ 0.01, *** = *p* ≤ 0.001, **** = *p* ≤ 0.0001. Error bars depict standard error.

**Figure 3 biology-12-00425-f003:**
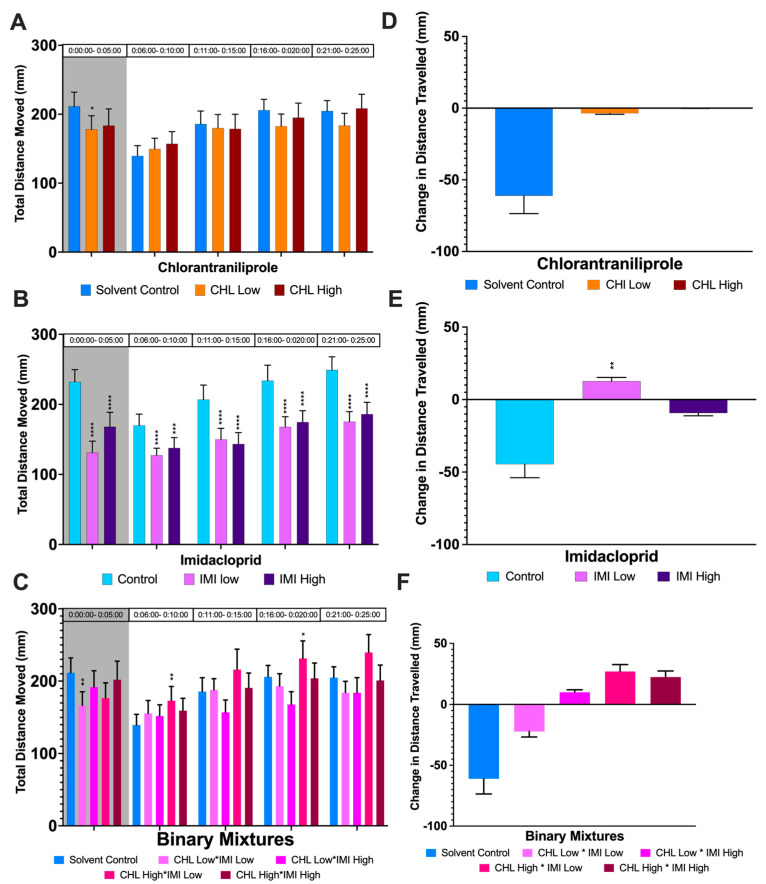
Effects of low (1.0 µg/L) and high (5.0 µg/L) concentrations of Chlorantraniliprole and Imidacloprid Single and Binary Mixtures on *D. magna* Swimming Behavior. (**A**–**C**) Average total distance moved (mm) of *D. magna* after 96 h of exposure to (**A**) CHL, (**B**) IMI, or (**C**) Binary Mixtures. The photoperiod for these was 10:20 min of dark:light (initial 5 min of acclimation not shown); (**D**–**F**) Photomotor response of *D. magna* after 96 h of CHL and IMI exposure, calculated by taking the difference between the average distance moved in the last minute of dark and the first minute of light. The significance of Tukey’s Multiple Comparisons tests is shown as * = *p* < 0.5, ** = *p* < 0.1, *** = *p* < 0.01, **** = *p* < 0.001. Error bars depict standard error.

**Table 1 biology-12-00425-t001:** Mortality of *D. magna* throughout a 96 h exposure to serial dilutions of water samples from 3 sites located throughout the Salinas River (CA, USA) collected September 2019 (before first flush). Significance of mortality relative to control group is shown as **** = *p* ≤ 0.0001.

		Percentage of Ambient Water					
	(100)	(60)	(35)	(20)	(12)	Control	
**Site ID**	**Percentage of *D. magna* Mortality**						**Toxicity**
Quail Creek	**100 ******	**100 ******	**100 ******	**100 ******	**87.5 ******	0	TOXIC
Alisal Creek	**100 ******	**100 ******	**62.5 ******	**27.5 ******	**15 ******	0	TOXIC
Salinas River	0	0	10	0	10	0	NON-OXIC

**Table 2 biology-12-00425-t002:** Mortality of *D. magna* throughout a 96 h exposure to serial dilutions of water samples from 3 sites located throughout the Salinas River (CA, USA) collected November 2019 (after first flush). Significance of mortality relative to control group is shown as *** = *p* ≤ 0.001, **** = *p* ≤ 0.0001.

	Percentage of Ambient Water					
	(100)	(20)	(12)	(6)	Control	
**Site ID**	**Percentage of *D. magna* Mortality**					**Toxicity**
Quail Creek	**100 ******	22.5	12.5	2.5	0	TOXIC
Alisal Creek	**100 ******	**50 *****	25	7.5	0	TOXIC
Salinas River	15	15	15	5	5	NON-TOXIC

## Data Availability

Analytical chemistry datasets referenced in this study were included in routine surface water monitoring performed by the California Department of Pesticide Regulation, Study 321 “Surface Water Monitoring for Pesticides in Agricultural Areas in the Central Coast and Southern California, 2019,” and are publicly archived at: https://www.cdpr.ca.gov/docs/emon/pubs/protocol/study321_monitoring_2019.pdf (accessed on 10 February 2023).

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
