# Peer review of "Swimming Behavior of Daphnia magna Is Altered by Pesticides of Concern, as Components of Agricultural Surface Water and in Acute Exposures"

_biology, 2023, doi:10.3390/biology12030425_

Round 1
Reviewer 1 Report
Manuscript entitled “Swimming behavior of Daphnia magna is altered by pesticides 2 of concern in agricultural surface water” needs major revision to be considered for publication.
The story is told chaotically and is incomplete. In my opinion it should be divided in to two parts. First analysis of water from sampling sites and second description of laboratory experiments. After such changes, the title should also be changed.
Results from first part determine further experiment this should be properly described. In material and method section the information’s are missing.
Authors diluted water samples: “…. we conducted a dilution series of our water samples to capture a wider range of toxic effects including mortality (lethal) and swimming behavior (sublethal) since organisms exposed to 100% water samples had high mortality.” What were the dilutions? – information given only in a table is not enough. What is high mortality?? - this is an exapmple
In results authors show LC50 values, there is no reference. Are these results obtained by authors – it is not said.
The reader is often forced to look for information between methods and results - this should not be the case. All information should be easily accessible and clearly presented.
Introduction is too long and given information is not that relevant for the study.
Sometimes authors put loosely related information into a single sentence. Such sentence needs rephrasing. E.g Line 33 “This research demonstrates that CHL, IMI, and contaminated surface waters all cause abnormal swimming behavior in D. magna, which are important prey for fish and also good bioindicators of effects on other zooplankton.”
Author Response
Reviewer 1 Comments and point-by-point responses
“The story is told chaotically and is incomplete. In my opinion it should be divided in to two parts. First analysis of water from sampling sites and second description of laboratory experiments. After such changes, the title should also be changed.”
Thank you for your comments and suggestions. We have extensively reorganized the manuscript to improve readability. Per Reviewer suggestions, we organized it into two parts: the analysis of agricultural surface water, followed by the laboratory experiments. We have also changed the title to better align with the updated organization of the manuscript (resubmitted manuscript, lines 2-4). The new title reflects the two main subdivisions of the study: the agricultural surface water exposures (before/after first flush) then the laboratory exposures (single/binary).
“Results from first part determine further experiment this should be properly described. In material and method section the information’s are missing. Authors diluted water samples: “.... we conducted a dilution series of our water samples to capture a wider range of toxic effects including mortality (lethal) and swimming behavior (sublethal) since organisms exposed to 100% water samples had high mortality.” What were the dilutions? – information given only in a table is not enough. What is high mortality?? - this is an example”
Thank you for the clarifying comments. In the Materials and Methods section we added a description of the dilution series for each sampling event: “For pre-first flush sampling, we used a dilution series of surface water concentrations: 100, 60%, 35%, 20% and 12% in order to observe a wide range of toxicological outcomes (resubmitted manuscript, lines 224-227)” and “For after first flush sampling, we used a broader dilution series: 100%, 30%, 20% 12%, and 6% in anticipation of higher chemical concentrations based on previous studies [46] (resubmitted manuscript, lines 232-233)” We also quantified high mortality in the context of the exposures (resubmitted manuscript, line 223. Dilutions for this study were determined based on high invertebrate mortality (97.5 – 100% mortality in undiluted water from two of the sampling sites) observed in previous exposure studies conducted at the same sites (Stinson et al. 2021). Additionally, we added additional details regarding how the dilutions were made (resubmitted manuscript, lines 227 -232).
“In results authors show LC50 values, there is no reference. Are these results obtained by authors – it is not said.”
Thank you for your comment. On line 313 of the resubmitted manuscript, where LC50 values are listed, we added the appropriate reference [43]; USEPA Aquatic Life Benchmarks and Ecological Risk Assessments for Registered Pesticides 2020. We used these LC50 values to inform our evaluation of whether a given measured concentration may be expected to result in behavioral change, based on established EC50/LC50 values (resubmitted manuscript, lines 240-241).
“The reader is often forced to look for information between methods and results - this should not be the case. All information should be easily accessible and clearly presented.”
Thank you for your suggestion to improve the accessibility of the Results section. We edited the description of the field sampling time points (“before” and “after”) in the Results to match the Methods section and improve clarity (resubmitted manuscript, lines 307, 326, 339, 356, 405). In the Results section we added references to the appropriate Methods subsection to reduce the need for the reader to search for information (resubmitted manuscript, lines 357, 374, 417-418).
“Introduction is too long and given information is not that relevant for the study.”
Thank you for the suggestion. We have removed several sentences from the introduction that may be less relevant to the study, in order to shorten and streamline the manuscript (resubmitted manuscript, lines 63-67, 85-86, 96-108, 120-121). Per the suggestion provided by Reviewer 2, we also added information about the role of D. magna in the aquatic food chain, the possibility of accumulation of pesticides in the tissues and any mechanisms of excretion (resubmitted manuscript, lines 109-116).
“Sometimes authors put loosely related information into a single sentence. Such sentence needs rephrasing. E.g Line 33 ‘This research demonstrates that CHL, IMI, and contaminated surface waters all cause abnormal swimming behavior in D. magna, which are important prey for fish and also good bioindicators of effects on other zooplankton.’“
Per the Reviewer’s suggestion, we have reviewed the manuscript to remove or edit sentences that contain unrelated or confusing information (resubmitted manuscript, lines 39-42, 72, 94, 463-464, 523, 596-597, 632-633). Thank you for the helpful suggestion to improve the readability of the manuscript. Included in our edits, we addressed the example sentence given on by removing the second half of the sentence entirely (resubmitted manuscript, lines 25-38).
Reviewer 2 Report
The paper titled “Swimming behaviour of Daphnia magna is altered by pesticides of concern in agricultural surface water”, reports important information related to the effects of pesticides on the swimming behaviour on this invertebrate model system.
The paper is very interesting because this model system represents a sensitive aquatic organism and because the use of pesticides represents a topic that arouses much concern and therefore it need new researches.
I suggest a minor revision
In the introductory part, the authors illustrate the potential of the experimental model but they should better specify the position of this organism in the food chain, the possibility of accumulation of pesticides in the tissues and any mechanisms of excretion. The relevant literature must be cited.
The authors should specify whether any recovery experiments have been done. This would allow to understand whether the induced effects were irreversible and would allow to obtain a complete profile of the effects of tested substances.
Finally I suggest to the authors to expand the conclusions that must not only contain a summary of the results but also a relationship with what is added to the information already present in the literature.
Author Response
Reviewer 2 Comments and point-by-point responses
“I suggest a minor revision
In the introductory part, the authors illustrate the potential of the experimental model but they should better specify the position of this organism in the food chain, the possibility of accumulation of pesticides in the tissues and any mechanisms of excretion. The relevant literature must be cited.”
Thank you for the suggestions. This information has been added to the Introduction (resubmitted manuscript, lines 109-116) along with relevant literature and will provide the reader with much-needed context. We kept this addition succinct because we received comments from another reviewer who recommended shortening the Introduction section.
“The authors should specify whether any recovery experiments have been done. This would allow to understand whether the induced effects were irreversible and would allow to obtain a complete profile of the effects of tested substances.”
Thank you for your comment. We did not include recovery experiments in the study due to time and space limitations but agree that this would be helpful for the reasons you listed. This is particularly relevant to our post first flush results from Quail Creek, which we suspect were indicative of delayed mortality (and of irreversible behavioral change). We have added text to the Discussion Section (resubmitted manuscript, lines 526-533) indicating that future studies should consider adding this as a component of their analyses.
“Finally I suggest to the authors to expand the conclusions that must not only contain a summary of the results but also a relationship with what is added to the information already present in the literature.”
To our knowledge, our study is the first to assess the behavioral responses of D. magna after exposure to binary mixtures of CHL and IMI (resubmitted manuscript, lines 615-617). Despite D. magna being used extensively in toxicology assessments, there are few publications examining the effects of environmental mixtures on their behavior. Thank you for pointing out that this was not clearly stated in the conclusions. We also explicitly stated that more research on the effects of mixtures is needed to understand potential impacts occurring in natural systems (resubmitted manuscript, lines 620-621, 629-630).
Round 2
Reviewer 1 Report
The last sentence of the abstract may be removed